# The Eco-Evo Mandala: Simplifying Bacterioplankton Complexity into Ecohealth Signatures

**DOI:** 10.3390/e23111471

**Published:** 2021-11-08

**Authors:** Elroy Galbraith, Matteo Convertino

**Affiliations:** 1Graduate School of Information Science and Technology, Hokkaido University, Sapporo 060-0814, Japan; 2bluEco Lab, Institute of Environment and Ecology, Tsinghua Shenzhen International Graduate School, Tsinghua University, Shenzhen 518055, China; matconv.uni@gmail.com

**Keywords:** marine microbiome, ecosystem health, biocomplexity, optimality, biogeochemical forcing, climate change, Mandala

## Abstract

The microbiome emits informative signals of biological organization and environmental pressure that aid ecosystem monitoring and prediction. Are the many signals reducible to a habitat-specific portfolio that characterizes ecosystem health? Does an optimally structured microbiome imply a resilient microbiome? To answer these questions, we applied our novel Eco-Evo Mandala to bacterioplankton data from four habitats within the Great Barrier Reef, to explore how patterns in community structure, function and genetics signal habitat-specific organization and departures from theoretical optimality. The Mandala revealed communities departing from optimality in habitat-specific ways, mostly along structural and functional traits related to bacterioplankton abundance and interaction distributions (reflected by ϵ and λ as power law and exponential distribution parameters), which are not linearly associated with each other. River and reef communities were similar in their relatively low abundance and interaction disorganization (low ϵ and λ) due to their protective structured habitats. On the contrary, lagoon and estuarine inshore reefs appeared the most disorganized due to the ocean temperature and biogeochemical stress. Phylogenetic distances (*D*) were minimally informative in characterizing bacterioplankton organization. However, dominant populations, such as Proteobacteria, Bacteroidetes, and Cyanobacteria, were largely responsible for community patterns, being generalists with a large functional gene repertoire (high *D*) that increases resilience. The relative balance of these populations was found to be habitat-specific and likely related to systemic environmental stress. The position on the Mandala along the three fundamental traits, as well as fluctuations in this ecological state, conveys information about the microbiome’s health (and likely ecosystem health considering bacteria-based multitrophic dependencies) as divergence from the expected relative optimality. The Eco-Evo Mandala emphasizes how habitat and the microbiome’s interaction network topology are first- and second-order factors for ecosystem health evaluation over taxonomic species richness. Unhealthy microbiome communities and unbalanced microbes are identified not by macroecological indicators but by mapping their impact on the collective proportion and distribution of interactions, which regulates the microbiome’s ecosystem function.

## 1. Introduction

### 1.1. Marine Ecosystem Health through the Microbiome

The Anthropocene presents an existential threat to many ecosystems, including marine habitats, necessitating ecological indicators that can assist in the prediction and management of environmental pressure. Global research on the Earth’s oceans [1,2,3,4,5] has revealed the large abundance, complexity, and ecological importance of microscopic organisms, which interact in communities called the microbiome. The microbiome is the ideal candidate for such indicators as they are largely responsible for the planet’s biogeochemical cycles [6,7,8] and are highly responsive to disturbances in these cycles [9,10]. Consequently, there is growing interest in the dynamics of microbiomes for indicating and predicting ecosystem health [11,12,13,14]. There is agreement that properly decoding marine microbiome signals (or those of any other aquatic ecosystem) is a critical first step in the assessment and consequent preservation and improvement of the resilience of ecosystems [15,16]. While there has been much success in the search for unique species (populations) predictive of ecosystem health states, more work is required to understand how the collective dynamics of the community can indicate ecosystem health, and there is value in understanding these dynamics within localities and regions [16], not merely at the global scale.

### 1.2. Leveraging Community Complexity for EcoHealth Assessment

Microbiomes, such as that found in the human gut [17], exhibit signatures of evolutionary optimality (or self-organizing criticality; see [18]) in which dynamical fluctuations are scale-free distributed, resulting in a relatively stable system. When these communities display a power law distribution in the frequency distribution of size classes among species (i.e., the abundance spectrum), their ecosystem is equally well-balanced (see [19] for plankton and [20] for oceanic biomass). Power law patterns have also been generally observed in scale-free networks [21] that were proven to coincide with a local or global energy minimum of system’s features. Consider river networks, whose power law distribution reflects the scale-free pattern of runoff and the energy minimum of water flow [22]. The stationary solution of the landscape evolution equation is a power law that is invariant across many orders of magnitudes (Rodriguez-Iturbe and Rinaldo, 2001); yet, it is linking patterns to processes clearly at stationarity. Invariance, which is a state of relative stability of a system’s features, is a byproduct of self-organized criticality but, more generally, of evolutionary dynamics leading to feasibly optimal states [23]. In this sense, some authors [24,25] have equated invariance with robustness, although the latter does not correspond to invariant features necessarily; for example, random networks are also topologically robust (yet invariant but not scale-invariant) because any perturbation leads to another random state.

In general, ecosystems may not be locally stable and undergo critical transitions (as in [26]) but these fluctuations are invariant over time. For instance, runoff in rivers may exhibit local instability (permanent due to structural features such as sudden jumps, or temporary such as black-swan variability due to climate extremes) but, over large spatial and temporal scales, shows stability in distribution. Deviation from global stability or invariance is a worrisome signature of departure from optimality—for instance, driven by large habitat and climatic modifications. A recent analysis of dynamical stability and invariance (which is indubitably dependent on the scale of analysis) has been performed for fisheries by [27]. Previous studies on the neural tissue [28] have also shown how scale-invariance is not only occurring at a critical transition, as in Bak (which is unstable), but at a global stable state, and this is the case for rivers. Lastly, probabilistically speaking, the stable distribution family is also sometimes referred to as the Lévy alpha-stable distribution to which power law distribution belongs. Thus, using this principle of relative optimality, one could evaluate the distribution of microbial species’ abundance and their interactions to assess the extent to which the microbiome is optimally arranged (and stable) or its divergence from the theoretically optimal state.

### 1.3. Community Signals as Environment-Modulated Noise

#### 1.3.1. Phylogenetic Trait Dynamics

The community dynamics of the marine microbiome contain many signals informative of environmental pressure. The color of this “noise”, in a dynamical sense, is extremely meaningful for the network topology underlying the observed ecological phenomena over time [27,29]. Traditional bioinformatics considers the dynamics of phylogenetic traits such as taxonomy or species abundance; for example, the diversity and abundances of populations within these communities change with habitat geomorphology and physicochemistry [30,31,32,33,34,35]. However, such approaches suffer from a lack of knowledge about many species [36]. Additionally, the taxonomic resolution at which such diversity analysis is conducted reveals other signals about the ecosystem state, because biological traits are differentially conserved along the phylogenetic tree [37]. Indeed, others have demonstrated how functional profiles are more informative than taxonomic profiles [38], given that environmental pressure can select for species with competitively advantageous functional potential.

#### 1.3.2. Entropic Methods for Species Interaction Assessment

There are also entropy-based approaches to species interaction assessment, which can aid ecosystem monitoring. The biomass conversion process of the system involves the interspecies abundance variation resulting from competitive (or cooperative) interactions, making information theoretic approaches, such as transfer entropy [39], suitable for signal detection. Transfer entropy (TE) is an assumption-free and probabilistic means of maximizing uncertainty reduction about species interactions [29,40]. It is ideal because species interactions are usually non-normal, non-linear, bidirectional, highly unpredictable, and asynchronous: methods such as TE have been found to be superior to correlational approaches [41,42], and other models for functional network inference [29]. Moreover, there is growing evidence of law-like behavior within communities, such as Taylor’s law and the abundance–size spectrum [19,43], which are highly robust and seem to shift only in the presence of significant environmental pressure. Ref. [17] showed that the information flow (measured by TE) within the gut microbiome of healthy individuals is more scale-free than that within the gut microbiome of unhealthy individuals. Therefore, here, we explore the usefulness of TE for ecosystem evaluation given its ability to handle the complexities within biosystems.

With so many signals to choose from, at so many (taxonomic and spatiotemporal) scales, it is daunting to condense the complexity of the microbiome into a simple yet useful portfolio of features that can inform ecosystem management. There is also a need to compare the usefulness of genetic and statistical features—such as information entropy—for ecosystem health evaluation. Furthermore, these signals can themselves be misleading. For example, one can conclude that a community is optimally structured because there is scale-freeness in its abundance-size spectrum, but this may not necessarily mean that the dynamical stability, its resilience, is high. The adaptability of microbes could lead to a re-structuring that presents as critical, although the rate of interaction among species has irreversibly transitioned, indicating low resilience.

### 1.4. Paper Outline

Thus, this paper is guided by three questions. First, are the collective dynamics of the marine microbiome reducible to a habitat-specific portfolio? Second, how informative are the genetic measurements for ecosystem health, compared to the statistical features, such as the distribution of abundance or information flow? Third, does an optimally structured microbiome, so characterized by its collective dynamics, imply a resilient microbiome? To answer the first, we disaggregate the microbiome into statistical distributions of its features—its genetic dissimilarity, interaction dynamics, and functional dynamics—and then model the relationship among them using our new Eco-Evo Mandala. This facilitates answering the second and third, because it enables exploration of the association between the structural stability and dynamic stability of the system, and comparison of the between-state differences along separate axes. This method is applied at the community level as a measure of habitat state, and at the population level to explore species-specific signals.

Using this approach, we explored the Great Barrier Reef, off the coast of Northern Australia. It is the world’s largest coral reef system, providing invaluable environmental, social, and economic services to humans and other species within this region. Though protected as a World Heritage Site, it is not spared the deleterious effects of anthropogenic perturbations such as agricultural runoff and climate change, which affect ecosystems at even the microbial scale [32,35,44]. The resultant coral bleaching events have been increasing in frequency and magnitude [45], necessitating robust ecosystem health evaluators such as the one presented here.

In Section 2 of this paper, we describe the materials and methods involved in this approach, using microbiome data collected from four geomorphologically different marine habitats within the Great Barrier Reef. We then present Section 3 and discuss the findings in Section 4, before concluding in Section 5.

## 2. Materials and Methods

### 2.1. Microbiome Data

Microbiome data initially published by [35] were used in this study. The data contained bacterioplankton abundances from 7 sites (TT1, TT2, TT3, TT4, FI, RI, and TR) along a 124 km transect of the Great Barrier Reef, measured monthly between 2009 and 2013 (see Figure 1). These sites represented 4 geomorphologically different aquatic habitats: estuarine inshore reef (TT1, TT2 and TT3), marine lagoon (TT4), marine inshore reef (FI and RI), and riverine (TR). The microbiological data contained 1140 operational taxonomic units (OTUs) of free-living marine bacterioplankton (bacteria and archaea kingdoms) obtained following 16s rRNA gene amplicon sequencing of water samples from each site. The taxonomic classifications of OTUs were derived using the Hitman bioinformatics workflow (see [35] for a description of the workflow and parameters used). Raw abundances based on sequence reads were used for analysis instead of relative species abundances. The number of water samples, and thus the time series lengths (*t*), differed for each site: *t* = 15 for TT1; *t* = 17 in TT2; *t* = 17 in TT3; *t* = 15 in TT4; *t* = 10 in FI; *t* = 9 in RI; and *t* = 3 in TR.

### 2.2. Interaction Quantification

Using $X=x_{1},x_{2},x_{3},\ldots ,x_{t}$ as the time series of abundances of length *t* for OTU *X*, the pairwise OTU interactions within each bacterioplankton community were quantified using transfer entropy (TE [17]):(1)$$\;TE_{X_{i}\to X_{j}}=\sum p\left(X_{j,t},X_{j,t-1},X_{i,t-1}\right)\cdot log_{2}\left(\frac{p(X_{j,t}|X_{j,t-1},X_{i,t-1})}{p(X_{j,t}|X_{j,t-1})}\right),$$
where $X_{i}$ and $X_{j}$ are the times series of bacterioplankton abundances for OTU *i* and *j*; and the subscript *t* (or t−1) indicates the value of $X_{i}$ and $X_{j}$ at time *t* (or t−1). TE was calculated using the box kernel estimator for continuous data from the JIDT Package [46], with a history length of 1 and kernel box size of 0.5 normalized units. TE quantifies the interdependence between OTUs as the information flux between them, considering the divergence, non-linearity, and asynchronicity in their abundance distribution, thereby reflecting predator–prey relationships.

### 2.3. Population Complexity Indicators

#### 2.3.1. Probabilistic Characterization of Distributions

The distribution of abundances and interactions (TEs) was characterized using one of two measures, λ or ϵ, derived from their exceedance probability distribution function (EPDF):(2)$$P\left(Y\geq y\right)∼\left\{\begin{array}{c} e^{-\lambda_{y}}\\ y^{-\epsilon +1} \end{array}\right.,$$
where *Y* was the abundances or interactions (see Section 2.2) at the population (taxonomic phylum) or community level; λ was used to characterize exponential distributions (see Appendix A); and Zipfian (power-law) distributions were characterized with ϵ (see Appendix A). The poweRlaw package [47] was used to estimate the parameters of all EPDFs using a maximum likelihood procedure to obtain the best fit for the distributions. Scale-freeness, and thus community or population optimality, increased with the value of ϵ or λ.

#### 2.3.2. Taylor’s Law

The variability of population abundances was characterized using the exponent ν of Taylor’s Law [48]:(3)$$\langle x^{2}\rangle ∼{\langle x\rangle }^{\nu },$$
where 〈x〉=E|X| is the mean abundance and $\langle x^{2}\rangle =E|X^{2}{\left|-E|X\right|}^{2}$ is the variance of the abundance. This exponent was estimated for each population at the phylum taxonomic level within each community, as well as for the entire community at each location. The parameter ν was estimated by determining the slope of a linear regression between the base 2 logarithm of both the population mean and variance (see Appendix A). Higher values of ν characterized communities or populations that experienced greater temporal instability in their abundance.

### 2.4. Phylogenetic Distance

The phyloseq package [49] was used to construct phylogenetic trees for each location’s community, considering the taxonomic classifications (rather than gene sequences) of all OTUs ever observed at a site (see Appendix A). Then, the pairwise distance between tips of the tree, which represented OTUs, was measured using the distTips function of the adephylo package [50]. This resulted in a matrix of phylogenetic distances between OTU pairs. The effective phylogenetic distance of a community *D* was calculated as the mean of these distances:(4)$$D=\frac{1}{n}\sum d_{i,j},$$
where *n* is the number of pairs in the distance matrix, and $d_{i,j}$ is the phylogenetic distance between OTU *i* and *j*. Communities with higher *D* have more genetic dissimilarity among their populations, which is related to functional dissimilarity. It should be noted that, typically, the phylogenetic distance is calculated as the sum of all pairwise distances between the tips of the tree in contrast to our effective average above. As each OTU was classified into a phylum *p*, the phylogenetic distance for each phylum within the phylogeny $D_{p}$ was calculated as the mean of all distances attributed to that phylum:(5)$$D_{p}=\frac{1}{n}\sum_{p}d_{i_{p},j},$$
where *n* is the number of pairs in the distance matrix and $d_{i_{p},j}$ is the phylogenetic distance between OTU *i* belonging to phylum *p* and all other OTUs *j* (see Appendix A). Phyla with higher $D_{p}$ are more genetically unrelated, or functionally dissimilar, to other phyla within their community.

### 2.5. Eco-Evo Mandala

Inspired by the work of [51] phytoplankton and [52] for microorganisms, we designed the Eco-Evo Mandala (see Figure 2A). Mandalas are visualizations of theoretical models or raw data of ecosystems along two or more dimensions and aim to (i) simplify observed complexities (in the physical or metaphysical planes) and (ii) detect ecosystem patterns [53]. Ref. [52] is a recent example of a “foraging Mandala” for aquatic microorganisms, where two axes were used to account for the local environment and individual biological adaptations (related to resource frequency and resource quality). Then, the Mandala allows us to reduce the vast biocomplexity of microbe–environment interactions into a pattern, where areas are distinct for a minimal number of fundamental parameters related to microbial strategies. This has also great relevance for the monitoring and management of aquatic environments by leveraging microbial information. In this study, the proposed Eco-Evo Mandala is a ternary plot that illustrates how a bacterioplankton community departs from optimality—in a habitat-specific way—given the relationship among its structure (ϵ as abundance distribution parameter; see Equation (Equation 2)), function (λ as interaction distribution parameter; see Equation (Equation 2)), and phylogenetic dissimilarity (*D* as effective phylogenetic distance; see Equation (Equation 4)). The values on each axis are rescaled to the maximum in each series (using $x/x_{max}$, where *x* is the value and $x_{max}$ is the maximum value observed). In the Mandala, ϵ and λ reflect more *Eco*logical processes (such as local speciation, long-range dispersal, and environment-driven interactions), while *D* reflects more long-term *Evo*lutionary processes (such as local genetic make-up through adaptation and selection). Communities departed from optimality the further they were plotted from the region reserved for high *D*, low λ, and low ϵ—in this case, the lower-left third that is the theoretically optimal ”healthy” state. Note that this healthy state is an ”absolute” optimal (considering high organization ϵ and λ, and effective functional diversity *D*); however, the habitat-specific optimal state may be located in another region of the Mandala, such as the ones we found for the observed habitats in this study.

## 3. Results

### 3.1. Community Health Characterization

We combined characterizations of community genetic relatedness, structure, and function into the Eco-Evo Mandala (see Figure 2) to evaluate ecosystem departure from optimal states. Genetic distance (*D*, see Section 2.4) measured the amount of genetic dissimilarity within a community (see Appendix A): high *D* meant that the community was very unrelated genetically (proportional to γ-diversity), while a low *D* meant that the community was highly related. The river mouth (TT1) had the most dissimilarity among its species (*D* = 6.9), followed by the estuarine inshore reef TT2 (*D* = 6.82); the lowest dissimilarity was observed in the river TR (*D* = 6.51). Structure was quantified as the amount of scale-freeness in the distribution of species’ abundances within a community (ϵ, see Section 2.3): lower values characterized more optimal structural organizations. The river (TR) was considered the most structurally optimal (ϵ = 1.74), followed by the river mouth (ϵ = 1.82), with the northernmost marine inshore reef FI showing the poorest structure (ϵ = 2.85). Function was characterized by the amount of scale-freeness in the distribution of species interactions (see Section 2.2) within a community: lower λ signaled more optimal distributions of community interactions. Again, the river was measured to be the most optimal (λ = 2.39), followed by the marine inshore reef RI (λ = 4.14); the estuarine inshore reef TT3 had the poorest functional organization (λ = 14.82). Sites plotted furthest from the region denoting high *D*, low ϵ, and low λ were considered furthest from optimality. Therefore, the Eco-Evo Mandala suggests that the river (TR) is the most optimal location within the region; the estuarine inshore reef TT3 was furthest from optimality for functional reasons; and the marine inshore reef FI was furthest from optimal for structural reasons.

By examining the pattern among points within the Mandala, we observed the relative importance of each signal and habitat class to the microbiome. We observed that the communities varied least along the genetic dissimilarity axis compared to the structural or functional axes of the Mandala (see Figure 2B). Genetic dissimilarity (*D*) ranged between 6.5 and 6.9 (sd = 0.125), while ϵ ranged between 1.74 and 2.85 (sd = 0.459), and λ ranged between 2.39 and 14.82 (sd = 4.102). We also observed greater between-habitat differences than within-habitat differences. The river (TR) was plotted furthest from all other points, the estuarine inshore reefs grouped together, and the marine sites, both lagoon (TT4) and the inshore reefs (FI and RI), were separated from the rest. Therefore, we concluded that the signals chosen for ecosystem monitoring were habitat-specific, and that genetic dissimilarity was the least significant of the three when evaluating ecosystem optimality, as communities are more likely to depart from optimality for structural or functional reasons.

### 3.2. Intra-Community Health Characterization

The phylum-level distribution of abundances best fit a Zipfian distribution (see Appendix A). The mean phylum-level parameters (|ϵ|) were highest in the marine inshore reef site RI (|ϵ| = 2.43), followed by the lagoon (TT4, |ϵ| = 2.35) and then the estuarine inshore reef site TT2 (|ϵ| = 2.06), least of all in the estuarine inshore reef site TT3 (|ϵ| = 1.72). In all estuarine inshore reefs and one marine inshore reef (FI), Euryarchaeota presented the lowest ϵ and was thus considered closest to optimal; in the other marine inshore reef (RI), Actinobacteria had the lowest; and for the marine lagoon, Bacteriodetes had the lowest. In the marine lagoon and one marine inshore reef (FI), the phylum with the highest ϵ, and therefore furthest from optimal, was Verrumicrobia, while in the other marine inshore reef (RI), the highest was Euryarchaeota. In the estuarine inshore reef designated TT1, the phylum with the highest exponent was Gemmatimonadetes; in TT2, it was Acidobacteria, and in TT3, it was Firmicutes (see Appendix A).

The phylum-level distribution of TE also best fit an exponential distribution (see Appendix A). The mean phylum-level parameters (|λ|) were highest in the estuarine inshore reef sites TT3, TT2, and TT1 (|λ| = 12.8, 10.6, and 10, respectively), followed by the marine inshore reef site FI (|λ| = 9.43), the marine lagoon (TT4, |λ| = 9.16), and then the marine inshore reef site RI (|λ| = 6.61). The phylum closest to interaction optimality (i.e., presenting the lowest λ) in the estuarine inshore reef TT1 was the Synergistes, while in TT2, it was Chlorofelxi, and in TT3, it was SBR1093; in the marine lagoon, Euryarchaeota had the lowest λ; for the marine inshore reef FI, the lowest λ was for Crearchaeota, while in RI, the lowest was for Verrumicrobia. The phylum furthest from optimal (that is, with the highest λ) in the marine inshore reef TT1 was Fibrobacteres, while in TT2 and TT3, it was WPS-2; in the marine lagoon, Spirochaetes had the highest λ; for the marine inshore reef FI, the highest λ was for Acidobacteria, while in RI, the highest was for Gemmatimonadetes (see Appendix A).

### 3.3. Community Structural and Functional Optimality

Another candidate metric for community and population characterization was ν, which described how the variance in the abundance of populations scaled with their mean abundance (see Equation (Equation 3)). At the community level, congruent with Taylor’s Law, ν ranged between 1.5 in the marine inshore reef FI and 1.7 in the river, approaching but not exceeding two (see Appendix A). Additionally, an inverse linear model provided a better fit between ν and ϵ than between ν and λ (see Figure 3). Note that linearity between ϵ and λ was also weak (see Figure 2B). However, at the population level, the pattern of the relationship among the metrics was much less clear. This notwithstanding, the top phyla considering ν and ϵ were comparable, as similar populations were shared between them (see Appendix A); contrarily, the top phyla considering λ (see Appendix A) were much less comparable to either of the other two. For example, in the marine inshore reef (FI), the same six Phyla were among the top 10 Phyla for ν and ϵ, but five were shared between λ and ϵ. Similarly, in the river (TR), more were shared between ν and ϵ than between λ and ν or ϵ. Consequently, while ν and ϵ appear to be encoding similar signals from the community, λ is measuring a different axis of variation.

## 4. Discussion

The research explored how the distribution of biological traits—genetic dissimilarity, abundance, and interactions—within marine bacterioplankton communities (and, in comparison, for a riverine community) could be used to evaluate ecosystem health. The bacterioplankton was considered due to the centrality of bacteria in ecosystem functions (e.g., carbon cycling) and multitrophic cascades with other species. Our findings evidenced: (1) the importance of probabilistic community organization indicators (distributions and their parameters) in contrast to population-threshold approaches alone (e.g., threshold on critical abundance of certain bacteria) that contain habitat-specific microbiome signals; (2) the sufficiency of abundance and abundance fluctuation information over time as eco-indicators of ecosystem health, without the need for further biological information; (3) the structural stability (of the physical habitat or community abundance proportion) does not necessarily imply dynamical stability that is largely dependent on long-term environmental fluctuation distribution affecting species interaction distribution.

### 4.1. Signaling Ecosystem Optimality with Community versus Population Dynamics

In this work, community diversity was quantified by *D*, the distribution of abundances by ϵ, and the distribution of functions by λ. Another candidate was ν; however, given the strong relationship between ν and ϵ, and that both ϵ and λ are derived from pdfs, we decided to construct the Mandala without ν. Previous work on the microbiome has concluded that community diversity, the distribution of abundances, and the distribution of functions represent separate axes of variation, each capable of signaling the ecosystem state. Note that here ”signaling” is purposely used in an information-theoretic perspective to underline the communication between species and environment manifested by time-series data. High *D* indicated high diversity within a community, which is considered ideal because it facilitates functional redundancy, and creates pools of dormant genes that can be used for atypical environmental conditions [38]. When highly diverse communities encounter stress, they can respond quickly by recruiting previously dormant populations who specialize in the stressed state. It also maintains community resilience despite population loss: functional redundancy ensures that necessary functions are maintained by allowing for population replacements. Low ϵ and low λ are considered optimal because they indicate that the distributions of abundances (structure) and interactions (function) are scale-free. Scale-free distributions, most notably in the abundance of biological species, have been associated with healthy biological systems [17,20]. The notion of self-organizing criticality [18] explains why this is so: biological systems tend to self-organize to optimal states, which manifest scale-freeness in the distribution of features [54]. We assume that an optimal habitat presents a diverse microbiome, with a scale-free distribution of functions and abundances. Therefore, habitats plotted in the portion of the Mandala reserved for high *D*, low λ, and low ϵ were considered closest to optimality.

This notwithstanding, these are theoretical expectations and we do not know the distribution or distribution parameter indicative of true optimality for the ocean bacterioplankton in relation to a habitat type. A distribution type and its parameter reflect a network; therefore, what we examine is variation in community assembly organizations that are likely evolutionarily optimal or departing from an optimal state. However, it is possible for the habitat microbiome to be reorganized into topologies different to scale-free networks. For instance, shallow lagoons without any major structural habitat forcing may lead to feasible optimality of the microbiome as distributed exponentially; these theoretically suboptimal conditions may also be related to recurrent biogeochemical loads. Consequently, we emphasize that our observations are likely about relative habitat optimality, and that future research is needed to identify the network topology of the bacterioplankton (and its distributions) associated with the optimal habitat baseline; this will allow us to carefully quantify departure from optimality due to diffuse and point-source stress such as climate oscillations (e.g., heatwaves) and nutrient loads.

Our Eco-Evo Mandala considers the distribution of traits within the entire community to evaluate the ecosystem state. When evaluating ecosystem health, one can focus on identifying keystone species [55], core to their ecological networks and whose exceedance of predefined thresholds destabilizes ecosystems. This justifies focusing on individual populations, because altering one can alter the entire system. However, while this implies the importance of the community interactions, it does not explicitly consider community dynamics; furthermore, it ignores the complexity of these community dynamics. The ecological function of the microbiome is the result of the interactions of all species within the meta-population, not any one in particular. This complexity means that the effects of one population can be offset by the collective dynamics of the entire system, rendering the information provided by a single population potentially misleading. Consequently, it is better to monitor entire communities when trying to evaluate an ecosystem, rather than specific populations. Here, we emphasize that community dynamics are not only informative of ecosystem health, but sometimes more so than population dynamics.

### 4.2. Phylogenetic and Probabilistic Ecohealth Characterization

It has been argued that the inclusion of biological knowledge can improve models of biosystem dynamics [56]. Phenotypic distributions hold information about the microbiome’s habitat [37], but care must be taken in selecting which traits to monitor: some, such as genetic dissimilarity, which contributes to diversity (richness), can be misleading. Additionally, the scale at which this analysis occurs, whether the population or community level, is important. Our results show that a community’s distributions of abundance and information flow are potentially more informative about the habitat than its diversity, similar to [57], who demonstrated how structural and dynamical indicators were sufficient for measuring system health. Although some studies have equated diversity with ecosystem health, some have found that diseased habitats can display higher microbial diversity than healthy habitats, particularly because of an increase in viruses [58,59,60]. We also demonstrated that analysis of the information entropy dynamics at the community scale—rather than at the population scale—could inform about the habitat state. This is important because there are many microbes that are as yet uncultured [61,62], so we do not have full taxonomic information about them, neither do we know their ecological function. Even if they were cultured, sometimes, in vitro behavior is different from in situ behavior [63]. Therefore, ecosystem health monitoring requires knowledge about the statistical dynamics of the entire community, rather than the taxonomic identity and ecological function of any particular species.

Interesting patterns emerged, which were consistent across ecological scales. First, the average behavior of populations (phyla) approximated the behavior of the entire community, resulting in the clustering of phyla around their habitats in the phase space Figure 4. Additionally, we observed novel inter-population relationships, which could be useful for ecosystem health evaluation. For example, the estuarine inshore reef TT3 presented Cyanobacteria, which were closer to optimality than Proteobacteria, and this community overall was the farthest of the seven sites from optimality. Conversely, the marine inshore reef RI presented Proteobacteria closer to optimality than Cyanobacteria, and this community overall was not too far from optimality. Cyanobacteria are widely implicated as problematic for marine ecosystems when they dominate microbial assemblages [64], while Proteobacteria are usually the dominant species characteristic of stable marine environments. Thus, it may be possible to discern the optimality of a habitat by comparing the structural and functional stability of select species.

### 4.3. Multi-Axis Community Departure from Optimality

One might assume that a structurally stable system is also dynamically stable, i.e., resilient. The results of the Mandala illustrate how a community can depart from optimal states for multiple reasons: optimality in one trait does not necessitate optimality in another. For example, the northern-most marine sites were closest to functional optimality but furthest from structural optimality, while the southern-most estuarine sites were closest to structural optimality but furthest from functional optimality. Exponents such as ϵ consider the stability of a community, reflecting the ability of the community structure to retain a particular shape given the probabilistic distribution of its abundances. Contrarily, λ, derived from transfer entropy, considers the dynamics of the community over time, the rate of energy sharing among populations. The fact that these biological systems display critical structures regardless of environmental state is a testament to their adaptive capabilities: regardless of the environmental state, the probability distribution of their abundances remains relatively constant. However, the community can still change in other ways: the taxonomic composition of these communities can transition following perturbation; the interaction rate among populations can also change. That is to say, structural stability does necessitate dynamical stability, as they represent separate axes. The results of this study caution researchers and environmental managers not to rely on solitary metrics to evaluate ecosystem health, but instead to confer with a combination of methods. Community optimality, as one indication of ecosystem health, can be inferred from the optimality in both structure and function represented by distributions in abundance and interactions, respectively. Another interesting finding was the hybrid nature of the distribution of TE in all habitats, which were mostly exponential but presented a scale-free tail. Microbiologists have theorized about the existence of a core microbiome, whose structure and function remain consistent regardless of geography, geomorphology, or environmental perturbation [65]. The core microbiome may be the portion of the interaction network manifesting this scale-free tail in the distribution. A network analysis approach may be able to confirm this by identifying whether hubs are present in the network graph following the application of a threshold to extract those nodes whose interactions are to be found at the tail of this distribution.

### 4.4. Habitat Inference from Bacterioplankton Organization

Many marine bacteria are ubiquitous, especially given the high motility of species and high porosity of habitats. This could mean that communities are roughly the same, rendering the marine microbiomes uninformative of habitat type or state, and thus only useful on a planetary scale. However, we observed greater between-class differences than within-class differences, supporting the observation of others that microbial communities collectively respond to the state and nature of their habitat in characteristic patterns [9,43,66,67,68]. We propose that the geomorphology, hydrodynamics, and water quality (temperature, salinity, nutrient loads, etc.) of these habitats may be responsible for the differentiation of microbial communities, which enables the identification of habitat class from community dynamics. Hydrodynamics coupled with eutrophication can increase microbial biomass in habitats [69], resulting in the relatively high abundance of particular species whose motility is positively impacted [70]. The river and estuarine sites present a shallower, more two-dimensional habitat constantly flushed by wave dynamics, which expedites the entry and exit of species and nutrients [71]. This may explain the comparatively low but exponentially distributed interactions of the estuarine sites, particularly the river mouth TT1, and the high yet less exponentially distributed interactions (indicative of a slower decay in the EPDF of interactions) of the marine inshore reef and marine lagoon. The low residence time of the water in the river and estuarine sites does not allow a stable environment for the build-up of interactions; on the other hand, the marine sites are less hydrodynamically turbulent and accessible from more angles, creating a calmer, more settled environment in which populations may grow and interact. The estuarine environment is more greatly impacted by local perturbation, such as industrial runoff from agricultural lands, which greatly increases nutrient loads, promoting growth—especially that of potentially toxic species—which may explain their less structured interactions and relatively stable abundances [35]. The marine sites are more impacted by global stressors such as temperature and salinity, with limited impact on nutrient availability [35], which would keep its interactions more stable. As such, fluctuations in structural stability and functional dynamics could be able to be indicative of the habitat geomorphology and environmental perturbation. This pattern would need to be explored further in other domains to confirm its validity.

We must mention that other ecological processes also impact community organization, not merely habitat geomorphology and environmental stress. Microbiome dynamics is also shaped by top-down factors, such as viral dynamics [72]. Recent studies focused on a complete characterization of ocean microbiome dynamics, i.e., the interaction of eukaryotes, prokaryotes, and viruses observed in several habitats around the world, to capture universal dynamical patterns. For instance, recently, Zhang et al. [73] considered sampled microbiomes in the Pearl River Delta and found associations with algal blooms and salinity gradients. Prodinger et al. [74] recently observed during diverse eukaryotic species blooms the Megavirus occurrence in the Uranouchi Inlet (JP), confirming the importance of viruses in the bacterioplankton dynamics. Debates still exist about the causal pathways between eukaryotes, prokaryotes, and viruses, and so further research is needed. Unfortunately, our study is limited by the lack of data on viral dynamics—and other biotic and abiotic factors, potentially—and does not allow us to quantify these other important features. However, the current study aims to characterize bacterioplankton patterns alone (due to the centrality of bacteria in many ecosystem functions and the data-driven approach) and not processes, based on the data available, along three main phenotypic axes reflecting abundance distribution, abundance-based interactions, and phylogenetics (eco and evo community traits). The model and the Mandala can, however, be extended to the whole microbiome (eukaryotic and viral communities too) rather than only being used for prokaryotes. We show that simply by considering abundance and interaction distribution at the community level, we are able to identify habitats and their potential relative stability or divergence from the theoretically optimal distribution. Additionally, we reiterate that phylogenetic dissimilarity did not help much with characterizing habitat patterns, although it may aid in understanding factors such as organization processes and effective diversity versus taxonomic diversity. By zooming out from the phylum to the OTU scale—where viral dynamics may appear evident—the uncertainty around habitat statistics, related to coupling abundance and interaction distribution parameters (ϵ and λ, respectively), increases because micro-determinants of populations become more important. Our Mandala is capable of signaling habitat-specific dynamics of bacterioplankton communities at the macro-level by using eco-evo traits, and this can also shed light on the importance of micro- versus macro-scale processes with relevance to ecological monitoring (i.e., what, where, and how much to sample bacterioplankton in relation to environmental dynamics).

## 5. Conclusions

In response to the question about how to characterize microbiome health, we found that the collective distribution of abundance and interactions is sufficient to assess the habitat-specific microbiome state, and even to predict habitat type from bacterioplankton organization. Biology (genetics) alone is not as informative for predicting an ecosystem state considering both statistical and entropic features. Additionally, we found that a structurally stable microbiome (considering abundance proportion related to static habitat features) does not equate to a dynamically stable one (dependent on microbial species interaction organization, which is sensitive to environmental fluctuations). Our conclusions are from analyses of distributions of community phenotypic traits of the bacterioplankton for four distinct aquatic habitats (one riverine and three marine). Traditional approaches to ecosystem diagnosis can miss habitat-specific features by focusing on abnormal readings in single species (or environmental drivers) rather than community patterns. Community patterns are, for instance, distribution functions of the collective organization of species (abundance, interaction, and phylogenetic dissimilarity from a distance/dissimilarity perspective, where the distance is associated with the autocorrelation function of these features), which may be independent of the spatial, temporal, and biological scale of organization. This invariance across scales is associated with stable or scale-free distribution, manifesting the relative optimality of a community, and the exponent of the distribution is likely habitat-specific (modulated by environmental dynamics) rather than blueprinted by biology. Thus, it is much more appropriate to focus on the collective distribution of abundance and interactions (as information flow due to the data-driven approach of ecosystem monitoring [29]) revealing ecosystem organization, rather than treating any specific bacterium and its abundance in isolation. By decoupling and decoding the genetic, structural, and functional signals emanating from aquatic microbiomes (here, focusing on the bacterioplankton but extendable to eukaryotic and viral components), we demonstrated a novel means of evaluating ecosystem health (as community organization). The results: (i) confirmed the relatively low importance of community genetic dissimilarity for ecosystem organization (and response, potentially); (ii) illustrated the complementarity but distinctness between structural and dynamic stability (biomass and interaction distributions, where the latter is much more sensitive to environmental fluctuations); and (iii) explored the habitat specificity of the community response to environmental stress by highlighting the important protective role of structural complexity, such as in marine reef and riverine habitats. In conclusion, it is possible—and preferable—to derive robust ecosystem optimality indicators from the information (probabilistic distributions) extracted from the biocomplexity of bacterioplankton communities.

## Figures and Tables

**Figure 1 entropy-23-01471-f001:**
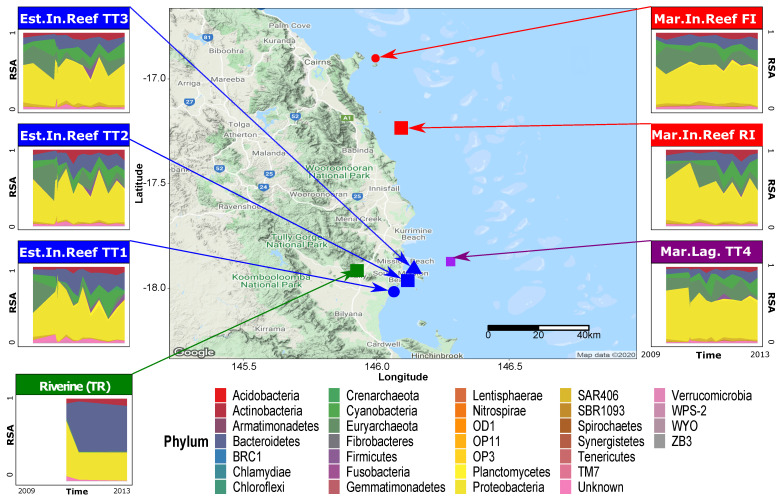
Site location and time series of relative abundance. The central panel is a Google Map of the 7 sites in the Great Barrier Reef used for this study. The surrounding panels present the time series of the relative abundance of phyla observed in each site. For the time series, the horizontal axis is the time series in chronological order, while the vertical axis is the relative abundance ranging from 0% to 100%. The color of the area in the time series plots corresponds with the phylum, as indicated in the legend below.

**Figure 2 entropy-23-01471-f002:**
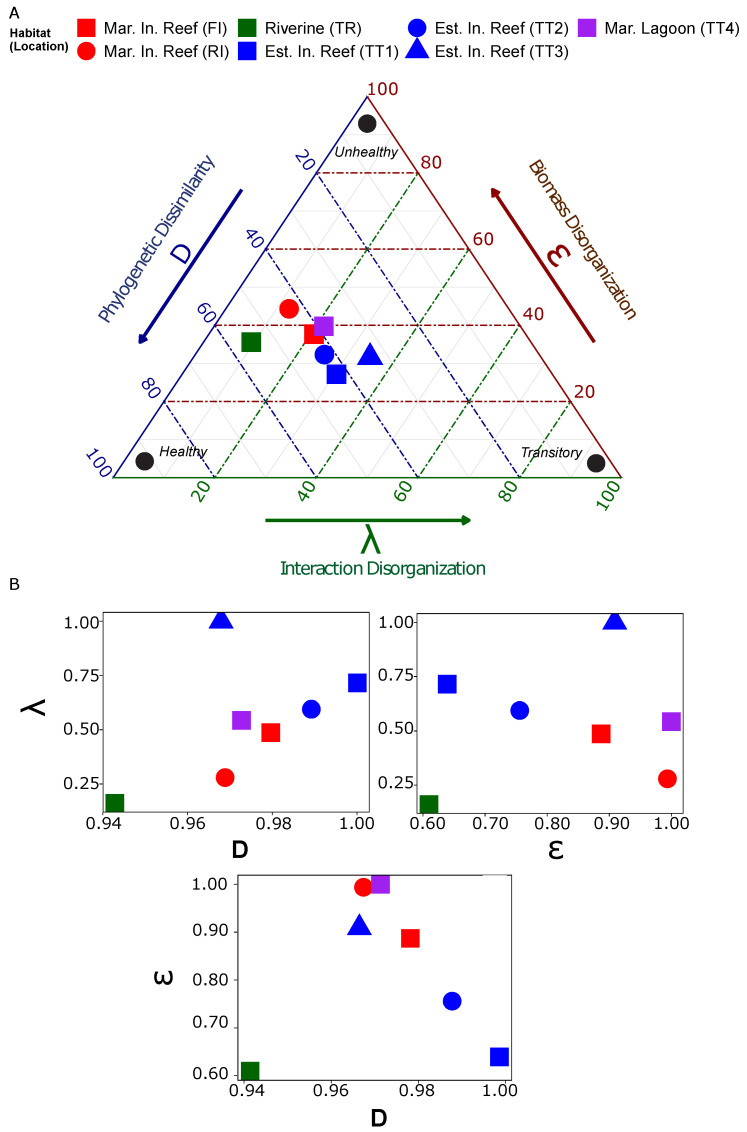
The Eco-Evo Mandala. (**A**). The Eco-Evo Mandala, signaling habitat departure from optimality considering the structure, function, and genetics of the microbiome. The Mandala is read counter-clockwise, the colored arrows exterior to the plot serving as guides: the blue axis is *D* measuring phylogenetic distance; the green axis is λ measuring the distribution of interactions; and the red axis is ϵ measuring the distribution of abundance. The colored grid lines indicate the position of points relative to the similarly colored axis (e.g., the blue grid lines show the position of a point relative to the *D* axis). *D* measures the amount of genetic dissimilarity within the network, ϵ measures the amount of structural organization, and λ measures the amount of organization in interactions among populations. (**B**) Pair-wise plots of the Mandala axes: D−λ, D−ϵ, and ϵ−λ. The community state on the Mandala along the three fundamental traits conveys information about the microbiome’s health (healthy, transitory, and unhealthy at the vertices as extreme states, and any other state) as divergence from the expected relative optimality.

**Figure 3 entropy-23-01471-f003:**
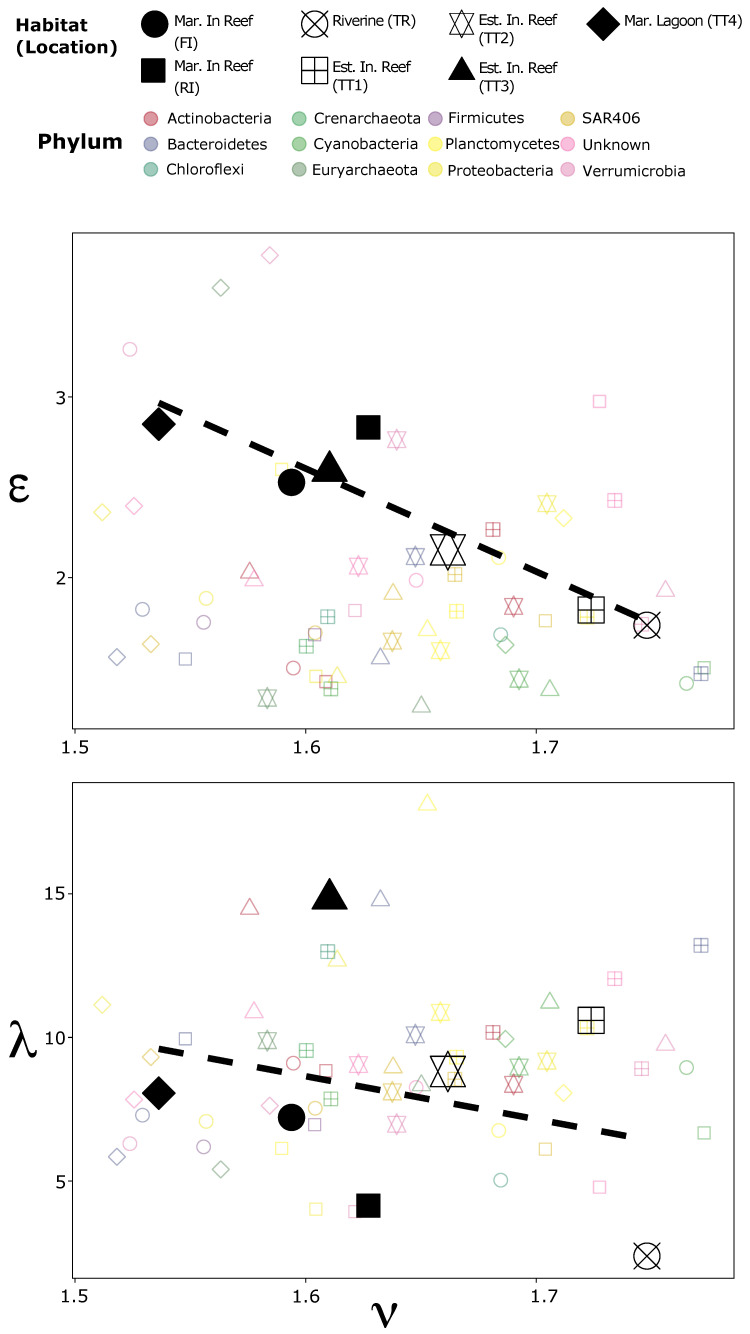
Associations among ν, ϵ, and λ. The shape of each point corresponds to the habitat (and location). The large black points are the community averages, while the smaller colored points indicate the values for specific phyla.

**Figure 4 entropy-23-01471-f004:**
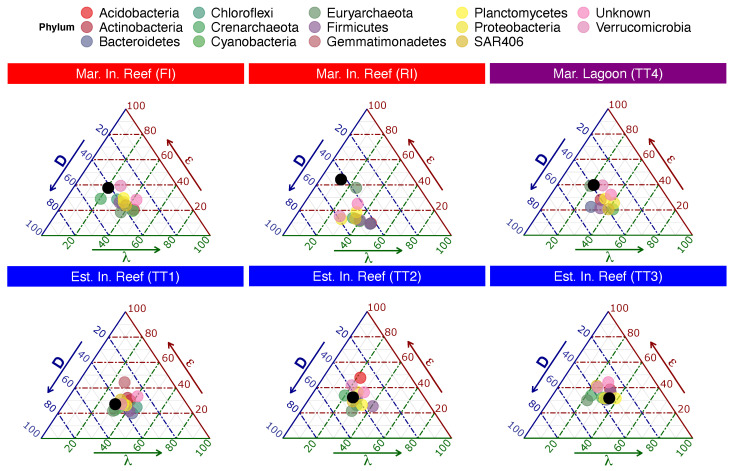
Phylum-level Mandala. Each panel is an Eco-Evo Mandala for the community sampled at each site. The colored points are the phyla while the black point is the community average. Only those phyla for which all three dimensions were calculable are displayed; the river (TR) is excluded as λ was not calculable for any of the populations (though it was for the entire population).

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
