# Peer review of "The Eco-Evo Mandala: Simplifying Bacterioplankton Complexity into Ecohealth Signatures"

_entropy, 2021, doi:10.3390/e23111471_

Round 1
Reviewer 1 Report
The manuscript is very interesting and well written and it is potentially very important for ecological studies, my only critics refers to the too sloppy and largely unmotivated conenction with Self-Organizing Criticality that has no necessary link with scale-free character of the networks, more specifically the entire statement: Biological complex systems like the microbiome exhibit self-organizing criticality (Bak et al., 1988) in which the system is stable and displays characteristic statistical features. For instance, in well balanced ecosystems, the frequency distribution of size classes among biota, the abundance-size spectrum, presents a power-law distribution (Cavender-Bares et al., 2001; Heneghan et al., 2019). A similar pattern has also been observed in scale-free networks (Barabasi and Bonabeau, 2003), largely considered the most stable systems, in which the distribution of the frequency of inter-node connections. largely considered the most stable systems, in which the distribution of the frequency of inter-node connections (interactions) follows a power-law distribution. Using this principle, one could evaluate the 63 distribution of the microbe abundances or their interaction (frequency or magnitude) to de64 termine the extent to which the microbiome is optimally arranged, a state expected to be 65 characteristic of an optimally arranged ecosystem should be eliminated, SOC refers to the presence of an 'attractor critical stet' in which systems stay on the 'edge of chaos' with continuous variation in response to continuous (and weak) environmental perturbations and this point is not demonstrated in the rest of the the work. The authors are suggested to refer to https://aip.scitation.org/doi/10.1063/5.0058511 and tohttps://www.researchgate.net/profile/Svetoslav-Nikolov-2/publication/6141237_Principal_difference_between_stability_and_structural_stability_robustness_as_used_in_systems_biology/links/54aa83800cf2bce6aa1d3c48/Principal-difference-between-stability-and-structural-stability-robustness-as-used-in-systems-biology.pdf for a thorough explanation of the meaning of SOC (and thus for its potential usefulness in their work).
If the authors will introduce these changes, in my opinion, the work is suited for publication on Entropy.
Author Response
Response to Reviewer 1 Comments
Reviewer 1
‘’The manuscript is very interesting and well written and it is potentially very important for ecological studies, my only critics refers to the too sloppy and largely unmotivated conenction with Self-Organizing Criticality that has no necessary link with scale-free character of the networks, more specifically the entire statement: Biological complex systems like the microbiome exhibit self-organizing criticality (Bak et al., 1988) in which the system is stable and displays characteristic statistical features. For instance, in well balanced ecosystems, the frequency distribution of size classes among biota, the abundance-size spectrum, presents a power-law distribution (Cavender-Bares et al., 2001; Heneghan et al., 2019). A similar pattern has also been observed in scale-free networks (Barabasi and Bonabeau, 2003), largely considered the most stable systems, in which the distribution of the frequency of inter-node connections. largely considered the most stable systems, in which the distribution of the frequency of inter-node connections (interactions) follows a power-law distribution. Using this principle, one could evaluate the 63 distribution of the microbe abundances or their interaction (frequency or magnitude) to de64 termine the extent to which the microbiome is optimally arranged, a state expected to be 65 characteristic of an optimally arranged ecosystem should be eliminated, SOC refers to the presence of an 'attractor critical stet' in which systems stay on the 'edge of chaos' with continuous variation in response to continuous (and weak) environmental perturbations and this point is not demonstrated in the rest of the the work. The authors are suggested to refer to https://aip.scitation.org/doi/10.1063/5.0058511 and to
https://www.researchgate.net/profile/Svetoslav-Nikolov-2/publication/6141237_Principal_difference_between_stability_and_structural_stability_robustness_as_used_in_systems_biology/links/54aa83800cf2bce6aa1d3c48/Principal-difference-between-stability-and-structural-stability-robustness-as-used-in-systems-biology.pdf for a thorough explanation of the meaning of SOC (and thus for its potential usefulness in their work).
If the authors will introduce these changes, in my opinion, the work is suited for publication on Entropy.’’
Response 1:
In relation to Section 1.2, Reviewer 1 believed our work’s connection with the concept of self-organizing criticality was not properly justified. We appreciate the comment and agree that this section would have confused the reader due to its poor writing. We consider the microbiome as a complex dynamical network that responds to environmental pressure by reorganizing itself into an evolutionarily optimal state (optimality should be considered in a relative sense rather than in an absolute sense). In a more general perspective, scale-free networks with power-law distributions, display relative/feasible and absolute energy minimum configurations after evolutionary path from random configurations. The signature of microbiome/network optimality is a byproduct of self-organizing criticality (if we consider SOC as one of the theory formulated in this area), but more generally a result of evolutionary dynamics that is common in nature considering both biotic and abiotic networks (for instance species interaction networks and river networks). We revised Section 1.2 (see lines 51-82) to focus the reader on the evolutionary dynamics which we leverage as an explanatory point to interpret our Mandala, and less on the concept of self-organizing criticality (that is a relatively discipline-specific theory). We draw on evidence from river systems to clarify the point. Additionally, we added a paragraph to Section 4.4 (see lines 317-324) to emphasize that we are yet to know the distribution type or parameter indicative of true evolutionary optimality of a particular habitat microbiome, as such our conclusions should be considering observations of relative habitat-optimality. The following sections has been added to the main paper
In Introduction:
‘’Microbiomes, like that found in the human gut (Li and Convertino, 2019), exhibit signatures of evolutionary optimality (or self-organizing criticality a la’ Bak et al. (1988)) in which dynamical fluctuations are scale-free distributed, resulting in a relatively stable system. When these communities display a power-law distribution in the frequency distribution of size classes among species (i.e. the abundance spectrum), their ecosystem is equally well-balanced (see Cavender-Bares et al. (2001) for plankton and Heneghan et al. (2019) for oceanic biomass). Power-law patterns have also been generally observed in scale-free networks (Barab ́asi and Bonabeau, 2003) that were proven to coincide with a local or global energy minimum of system’s features. Consider river networks, whose power-law distribution reflects the scale- free pattern of runoff and the energy minimum of water flow (Rodriguez-Iturbe and Rinaldo, 2001). The stationary solution of the landscape evolution equation is a power-law that is invariant across many orders of magnitudes (Rodriguez-Iturbe and Rinaldo, 2001); yet, it is linking patterns to processes clearly at stationarity. Invariance, that is a state of relative stability of system’s features, is a byproduct of self-organized criticality, but more generally of evolutionary dynamics leading to feasibly optimal states (Hidalgo et al., 2014). In this sense some authors (Zimatore et al., 2021; Nikolov et al., 2007) have called invariance as robustness, although the latter does not correspond to invariant features necessarily: for example, random networks are also topologically robust (yet invariant but not scale-invariant) because any perturbation leads to another random state.
In general, ecosystems may not be locally stable and undergo critical transitions (a la’ Scheffer et al. (2009)) but these fluctuations are invariant over time. For instance runoff in rivers may exhibit local instability (permanent due to structural features such as sudden jumps, or temporary such as black-swan variability due to climate extremes) but over large spatial and temporal scales shows stability in distribution. Deviation from global stability or invariance is a worrisome signature of departure from optimality, for instance, driven by large habitat and climatic modifications. A recent analysis of dynamical stability and invariance (that is indubitably dependent on the scale of analysis) has been done for fisheries by Li and Convertino (2021b). Previous studies on the neural tissue (Martinello et al., 2017) have also shown how scale-invariance is not only occurring at a critical transition a la’ Bak (that is unstable) but at a global stable state, and this is the case of rivers.
Lastly, probabilistically speaking, the stable distribution family is also sometimes referred to as the L ́evy alpha-stable distribution to which power-law distribution belong. Thus, using this principle of relative optimality, one could evaluate the distribution of microbial species abundance and their interactions to assess the extent to which the microbiome is optimally arranged (and stable) or its divergence from the theoretically optimal state.‘’
In Discussion:
‘’This notwithstanding, these are theoretical expectations and that we do not know the distribution or distribution parameter indicative of true optimality for the ocean bacterioplankton in relation to a habitat type. A distribution type and its parameter reflect a network, therefore what we examine is variation in community assembly organizations that are likely evolutionarily optimal or departing from an optimal state. However, It is possible for the habitat microbiome to be reorganized into topologies different than scale-free networks. For instance, shallow lagoons without any major structural habitat forcing may lead to feasible optimality of the microbiome as distributed exponentially; this theoretically suboptimal conditions may also be related to recurrent biogeochemical loads. Consequently, we emphasize that our observations are likely about relative habitat optimality, and that future research is needed to identify the network topology of the bacterioplankton (and its distributions) associated with optimal habitat baseline; this will allow to carefully quantify departure from optimality due to diffuse and point-source stress such as climate oscillations (e.g. heatwaves) and nutrient loads.’’
Reviewer 2 Report
The manuscript by Galbraith and Convertino have suggested the microbiome’s interaction network topology to be more important for ecosystem evaluation than taxonomic considerations by applying their new Eco-Eco Mandala model. Moreover, they combined characterizations of community genetic relatedness, structure, and function into the Eco-Evo Mandala (see Fig.2) to evaluate ecosystem departure from optimal states. This manuscript is very interesting and refreshing. Overall the manuscript is nicely written and is a subject of interest to aquatic microbial ecologists in today’s context. I have some minor comments and hope the authors would incorporate this in the manuscript
L70-72 & L393-395: In addition to habitat geomorphology and physicochemistry, top-down factors especially viral lysis also in number of studies have shown to influence and tend to strongly impact bacterial abundance and diversity. While bacterial community composition is bottom-up controlled, bacterial diversity is largely impacted by viral activity. Moreover, in eutrophic environments the impact of viruses has been shown to have large impact on bacterial diversity indices.
Figures2,3&4: In both X and Y axis, the font size is too small to read.
Author Response
Response to Reviewer 2 Comments
Point 1:
‘’The manuscript by Galbraith and Convertino have suggested the microbiome’s interaction network topology to be more important for ecosystem evaluation than taxonomic considerations by applying their new Eco-Eco Mandala model. Moreover, they combined characterizations of community genetic relatedness, structure, and function into the Eco-Evo Mandala (see Fig.2) to evaluate ecosystem departure from optimal states. This manuscript is very interesting and refreshing. Overall the manuscript is nicely written and is a subject of interest to aquatic microbial ecologists in today’s context. I have some minor comments and hope the authors would incorporate this in the manuscript
L70-72 & L393-395: In addition to habitat geomorphology and physicochemistry, top-down factors especially viral lysis also in number of studies have shown to influence and tend to strongly impact bacterial abundance and diversity. While bacterial community composition is bottom-up controlled, bacterial diversity is largely impacted by viral activity. Moreover, in eutrophic environments the impact of viruses has been shown to have large impact on bacterial diversity indices.
Response 1:
In relation to Section 4.4, Reviewer 2 noted that we did not mention top-down factors (played out by small-scale organisms that are not bacteria) which can impact community dynamics, particularly viral lysis. We thank the reviewer and agree that there are other factors at work underneath the dynamics we studied. Unfortunately, our data did not allow us to comment on these other factors despite their legitimacy, so we felt it improper to discuss them. This notwithstanding, our purpose was not to elucidate the processes involved in community dynamics, but to explore the macro-scale patterns considering bacterioplankton data (that are becoming more and more available), and how these differ between habitat-types. We emphasize macro-scale patterns, because drilling down to the OTU level (e.g. to explore viral lysis) would potentially render habitat-level patterns difficult to see. We in fact observed that the more we zoom down into the biological scales of organization (e.g. from phyla to species) the uncertainty around habitat-scale average increases (as expected). We added a paragraph to Section 4.4 to remind the reader that other factors impact community dynamics but, while our work is not ignoring them, we are focused on macro-scale patterns considering bacterioplankton organization along three ‘’phenotypic’’ axes. (See lines 444-456)
We must mention that other ecological processes also impact community organization, not just habitat geomorphology and environmental stress. Microbiome dynamics is also shaped by top-down factors, such as viral dynamics (Chow et al., 2014). Recent studies focused on a complete characterization of ocean microbiome dynamics, that is the interaction of eukaryotes, prokaryotes, and viruses observed in several habitats around the world, to capture universal dynamical patterns For instance, Zhang et al. (2021) considered sampled microbiomes in the Pearl River Delta and found associations with algal blooms and salinity gradients. Prodinger et al. (2021) recently observed during diverse species blooms with Megavirus occurrence in the Uranouchi Inlet (JP), confirming the importance of viruses into the bacterioplankton dynamics. Debates still exist about the causal pathways between eukaryotes, prokaryotes, and viruses, and so further research is needed.
Unfortunately, our study is limited by the lack of data on viral dynamics - and other biotic and abiotic factors, potentially - and does not allow us to quantify these other important features. However, the current study aims to characterize bacterioplankton patterns alone (due to the centrality of bacteria in many ecosystem functions and the data-driven approach) and not processes, based on data available, along three main phenotypic axes reflecting abundance distribution, abundance-based interactions and phylogenetics (eco and evo community traits). The model and the Mandala can however be extended to the whole microbiome (eukaryotic and viral communities too) rather than just used for prokaryotes. We show that just by considering abundance and interaction distribution at the community level we are able to identify habitats and their potential relative stability or divergence from the theoretically optimal distribution. Additionally, we reiterate that phylogenetic dissimilarity did not help much with characterizing habitat patterns, although it may aid in understanding factors organization processes and effective diversity versus taxonomic diversity. By zooming down from the phylum to the OTU scale -- where viral dynamics may appear evident – the uncertainty around habitat statistics, related to coupling abundance and interaction distribution parameters (epsilon and lambda, respectively), increases because micro-determinants of populations become more important.
Our Mandala is capable of signaling habitat-specific dynamics of bacterioplankton communities at the macro-level by using eco-evo traits, and that can also shed light into the importance of micro- versus macro-scale processes with relevance to ecological monitoring (i.e. what, where and how much to sample bacterioplankton in relation to environmental dynamics).
Point 2:
Figures2,3&4: In both X and Y axis, the font size is too small to read.’‘’
Response 2:
Reviews 2 pointed to other editorial and stylistic issues to be addressed, including typographical errors and illegible text in figures. We deeply appreciate the time the reviewer took to point these out to us. We have worked on the main figures to improve legibility and corrected typographical errors in the text.
Reviewer 3 Report
The work is a multivariate analysis of the dependences in the distributions of various populations of marine bacterioplankton in the Great Barrier Reef in Australia. The authors carried out correct collection of samples and revealed phylogenetic diversity at several experimental points. The authors use statistical approaches correctly. In addition, the authors acknowledge the complexity of both the organization of bacterioplankton communities and the complexity of describing the dependences of its distribution. The approach used is a multi-plane analysis of these distributions at a different level from the species and its abundance in the community to the distribution of higher taxa, such as Philae and Empires. The analysis was supposed to identify those dependencies and relationships in distributions that show the health of the ecosystem. The illustrative material shows the possibility of identifying the organization of bacterioplankton communities when it can be correlated with the health of the marine ecosystem. At the same time, the authors acknowledge the complexity and multidimensionality of such an analysis. Interesting conclusions that the conclusion on the abundance of abundance is more informative about the health of the ecosystem, does not detract from the significance in the distribution of higher taxa. The article is provided with an essential bibliographic apparatus, references in the text are cited in the list and vice versa. The article is interesting, based on sufficient experimental material with conclusions that coincide with the tasks set, and can be published in Entropy journal with minor revision, a list of corrections is given below. Particular attention should be paid to the spaces between the individual elements of the text, Capital letters in the names of higher taxa, and also to the design of the reference list.
Line 13 – bcomplexity = mistake?
Line 137 - 124km = space
Line 166 - Fig.S7A and Fig.S5A = spaces
Line 177 - Fig.S6A = space
Line 182 - Fig.S8 = space
Lines 190 and 191 – phylum = must be Phylum
Line 193 – phylum = must be Phylum; Fig.S8 = space
Line 201 - eq.2.2 = space
Line 202 - eq.2.4 = space
Line 209 - Fig.2 = space
Line 210 - Sec.2.4 = space
Line 211 - Fig.S8 = space
Line 216 - Sec.2.3 = space
Line 220 – Sec.2.2 = space
Line 231 – Fig.2B = space
Line 241 – phylum = must be Phylum; Fig.S5A = space
Line 242 – phylum = Phylum
Lines 248, 250 – phylum = Phylum
Line 251 – Fig.S5B = space
Lines 253, 254, 257, 261 – phylum = Phylum; Fig.S7A = space
Line 264 - Fig.S7B = space
Line 268 - Sec.2.3 = space
Line 269 - Fig.S6 = space
Line 271 - Fig.2B = space
Line 273 – phyla = Phyla
Line 274 - S.S5B = space
Line 275 - Fig.S7B = space
Line 276 - 6 phyla, 10 phyla = 6 Phyla, 10 Phyla
Line 309 - $epsilon = check it
Line 344, 345 – phyla = Phyla
Line 445 - Msystems, = mSystems,
Line 448 - Applied microbiology and biotechnology = Applied Microbiology and Biotechnology
Line 450 - Physical review = Physical Review
Line 455 - Scientific American = Scientific American
Line 465 - PloS one = PLOS One
Line 472, 570, 579 - Frontiers in microbiology = Frontiers in Microbiology
Line 475 - The ISME journal = The ISME Journal
Line 480 - MSystems, = mSystems,
Lines 483, 526, 549 - Frontiers in microbiology = Frontiers in microbiology
Line 489 - science, = Science,
Line 497 - Communications biology = Communications Biology
Line 542, 620 - Marine pollution bulletin = Marine Pollution Bulletin
Line 546 – Scientific reports = Scientific Reports
Line 559 - Oceanologica acta = Oceanologica Acta
Line 589 - Physical review letters = Physical Review Letters
Line 600 - Molecular ecology = Molecular Ecology
Line 611 - Environmental microbiology = Environmental Microbiology
Line 617 - Nature communications = Nature Communications
Line 642, 645 – phylum = Phylum
Line 656, 661, 664, 672, 676 – phyla = Phyla
Author Response
Response to Reviewer 3 Comments
Point 1:
‘’The work is a multivariate analysis of the dependences in the distributions of various populations of marine bacterioplankton in the Great Barrier Reef in Australia. The authors carried out correct collection of samples and revealed phylogenetic diversity at several experimental points. The authors use statistical approaches correctly. In addition, the authors acknowledge the complexity of both the organization of bacterioplankton communities and the complexity of describing the dependences of its distribution. The approach used is a multi-plane analysis of these distributions at a different level from the species and its abundance in the community to the distribution of higher taxa, such as Philae and Empires. The analysis was supposed to identify those dependencies and relationships in distributions that show the health of the ecosystem. The illustrative material shows the possibility of identifying the organization of bacterioplankton communities when it can be correlated with the health of the marine ecosystem. At the same time, the authors acknowledge the complexity and multidimensionality of such an analysis. Interesting conclusions that the conclusion on the abundance of abundance is more informative about the health of the ecosystem, does not detract from the significance in the distribution of higher taxa. The article is provided with an essential bibliographic apparatus, references in the text are cited in the list and vice versa. The article is interesting, based on sufficient experimental material with conclusions that coincide with the tasks set, and can be published in Entropy journal with minor revision, a list of corrections is given below. Particular attention should be paid to the spaces between the individual elements of the text, Capital letters in the names of higher taxa, and also to the design of the reference list.’’
Response 1: We gratefully thank Reviewer #3 for the very positive comments and understanding of our paper. The reviewer pointed to other editorial and stylistic issues to be addressed, including typographical errors and illegible text in figures. We deeply appreciate the time the reviewer took to point these out to us. We have worked on the main figures to improve legibility and corrected typographical errors in the text.
Round 2
Reviewer 1 Report
The authors met all my requirements and now the manuscript is better 'tuned' in terms of the theoretical claims.
Reviewer 2 Report
I am satisfied with the revisions.
This manuscript is a resubmission of an earlier submission. The following is a list of the peer review reports and author responses from that submission.